# Plant Nanovesicles for Essential Oil Delivery

**DOI:** 10.3390/pharmaceutics14122581

**Published:** 2022-11-24

**Authors:** Mónica Zuzarte, Carla Vitorino, Lígia Salgueiro, Henrique Girão

**Affiliations:** 1Coimbra Institute for Clinical and Biomedical Research (iCBR), Faculty of Medicine, University of Coimbra, 3004-548 Coimbra, Portugal; 2Center for Innovative Biomedicine and Biotechnology (CIBB), University of Coimbra, 3004-548 Coimbra, Portugal; 3Clinical Academic Centre of Coimbra (CACC), 3004-548 Coimbra, Portugal; 4Faculty of Pharmacy, University of Coimbra, 3004-548 Coimbra, Portugal; 5Chemical Process Engineering and Forest Products Research Centre (CIEPQPF), Department of Chemical Engineering, Faculty of Sciences and Technology, University of Coimbra, 3030-790 Coimbra, Portugal

**Keywords:** volatile compounds, plant-derived vesicles, extracellular vesicles, functionalized vesicles

## Abstract

Essential oils’ therapeutic potential is highly recognized, with many applications rising due to reported anti-inflammatory, cardioprotective, neuroprotective, anti-aging, and anti-cancer effects. Nevertheless, clinical translation still remains a challenge, mainly due to essential oils’ volatility and low water solubility and stability. The present review gathers relevant information and postulates on the potential application of plant nanovesicles to effectively deliver essential oils to target organs. Indeed, plant nanovesicles are emerging as alternatives to mammalian vesicles and synthetic carriers due to their safety, stability, non-toxicity, and low immunogenicity. Moreover, they can be produced on a large scale from various plant parts, enabling an easier, more rapid, and less costly industrial application that could add value to waste products and boost the circular economy. Importantly, the use of plant nanovesicles as delivery platforms could increase essential oils’ bioavailability and improve chemical stability while reducing volatility and toxicity issues. Additionally, using targeting strategies, essential oils’ selectivity, drug delivery, and efficacy could be improved, ultimately leading to dose reduction and patient compliance. Bearing this in mind, information on current pharmaceutical technologies available to enable distinct routes of administration of loaded vesicles is also discussed.

## 1. Current Overview of Plant Nanovesicles

The field of plant nanovesicles has recently emerged and is evolving quite rapidly, with the number of publications growing exponentially in the last year [1]. Since interspecies cell-to-cell communication through nanoparticles has been recognized, the interest in this field has been boosted and has attracted members of the scientific community from diverse areas. Given their interdisciplinary nature and clinical potential, these vesicles are currently in the spotlight as new biologically active substances or natural delivery platforms. Despite this advance, plant nanovesicles categorization remains confusing and lacks consensus. Indeed, the term plant-derived nanovesicles (PDNVs) has been indistinctly used to define extracellular vesicles (exosomes), intracellular vesicles existing in plants, artificial vesicles formed during extraction processes, or even natural vesicle mimics prepared from plant-derived molecules [2]. As a matter of fact, establishing an accurate nomenclature for plant vesicles is quite challenging due to uncertainties about their origin, lack of standardized isolation methods, and the puzzling piece of how the cell wall barrier is overcome [3]. Moreover, differences in size range and molecular composition are quite evident. Overall, only a limited number of studies refer to vesicles purified from the apoplastic fluid (extracellular vesicles), whereas the majority resort to blending and juicing processes that, besides isolating extracellular vesicles, also recover intracellular vesicles, artificial nanoparticles, and microsomes formed by ‘re-mixing’ of disrupted cellular membranes formed during the isolation procedures [4]. For these cases, the general term plant-derived nanovesicles (PDNVs) or exosome-like nanoparticles (ELNs) is generally used, whereas apoplastic vesicles, which represent purer fractions, are referred to as plant extracellular vesicles (EVs) [5]. EVs are associated with Unconventional Protein Secretion (UPS) routes, and although their exact secretion pathway remains unclear, possibilities include exocyst-positive organelle-mediated secretion (EXPO), vesicle budding from the plasma membrane, and multivesicular body fusion with the plasma membrane. Indeed, three main classes of plant EVs have been identified: (1) EXPO vesicles varying between 20 nm and 500 nm that are released into the apoplastic fluid and can be recognized using the protein maker Exo70E2; (2) microvesicles with sizes ranging from 150 nm to 1 µm that bud off the plasma membrane and seem to be positive for syntaxin SYP121, often referred to as PEN1; (3) exosomes which present smaller sizes (30–150 nm) and specifically contain the tetraspanin protein (TET8), as reviewed by Farley and colleagues [6]. More recently, EVs derived from plant exudates such as sap, gum, resins, and root exudates are also being investigated and are emerging as a very appealing research field [7,8,9]. Importantly, as blending and juicing processes are easy to apply and enable higher vesicle yield, they will likely continue to be used in future studies, thus disregarding vesicle subcellular origin [1].

Plant nanovesicles are generally obtained from edible plants such as ginger (*Zingiber* sp.), grapes (*Vitis* sp.), lemons, and grapefruits (*Citrus* sp.), which enables a large-scale, more rapid, and less costly industrial application [10]. Furthermore, these vesicles can be obtained from various plant parts, including roots and dried parts, which could add value to waste products in agriculture, thus promoting a circular economy and contributing to more eco-sustainable productions.

Plant nanovesicles carry a wide variety of molecules, including proteins, lipids, miRNAs, vitamins, and plant metabolites (e.g., gingerol, sulphorane, and naringenin), with some differences in comparison to mammalian EVs, as shown in Figure 1 [11].

In addition to their relevant role in plant growth, defense, and symbiotic relations [12], many studies suggest that plant nanovesicles are able to transfer their own biological cargo, thus impacting recipient cells, including those from different species. Indeed, several in vivo studies have already validated the bioactive potential of these vesicles. For example, PDNVS from grapes, grapefruit, and ginger have shown anti-inflammatory properties, with beneficial effects observed in intestinal homeostasis and repair mechanisms [13,14,15], and those from citric fruits, such as lemon and grapefruit, have been pointed out for their antineoplastic effects [16,17]. As extensive reviews on the potential therapeutic effects of plant nanovesicles have already been conducted by others [1,2], they will not be herein detailed. Regarding clinical trials, only a few have been carried out, as also discussed by others [18]. Furthermore, several patents on plant vesicles have been disclosed. Although the majority refer to production processes, a few focus on the therapeutic potential of these vesicles, as exemplified in Table 1.

In addition to plant nanovesicles’ bioactive potential and their use in preventive and therapeutic approaches, these vesicles can also be explored as drug-delivery platforms. In fact, plant nanovesicles are emerging as promising alternatives to mammalian nanocarriers due to their safety, stability, non-toxicity, and low immunogenicity [19]. Moreover, many mammalian EVs are involved in tumor biology, and their biohazard potential remains unclear [1], thus reinforcing the interest in nanovesicles of plant origin. Nevertheless, it has been suggested that for a better-targeted delivery, nanovesicles should be offloaded of their own cargo to enable a higher load efficiency of the compounds of interest [20]. Also, the extraction of plant nanovesicle-derived lipids and their reassembly has been suggested to obtain more uniform-sized vesicles [21]. Regarding cargo loading, both cell-based and non-cell-based approaches can be used, as recently overviewed by Han and colleagues [22]. Briefly, in the first, donor cells are manipulated, and cargo loading occurs before vesicle isolation. On the other hand, using non-cell-based approaches, cargo is loaded after isolation through passive (diffusion) or active loading strategies (electroporation or sonication) [22]. Despite the scientific evidence, plant nanovesicles remain underutilized as therapeutic modalities, and the development of marketable products still needs to take into account several limiting factors. Indeed, while characterization methods follow those established for mammalian EVs (dynamic light scattering, nanoparticle analysis, electron microscopy, PCR, Western blot, mass spectrometry, and sequencing), separation techniques require standardization to fit mass production. In addition, the purification of appropriate vesicles for specific uses still remains a challenge. Indeed, for EVs alone, several techniques with distinct drawbacks are available, such as differential ultracentrifugation, density gradient, PEG precipitation, and size exclusion chromatography. For example, using ultracentrifugation, inconsistencies due to centrifugation speed, force, and type of rotor have been reported, and co-contamination with proteins [23] is frequent. Also, density gradients are quite time-consuming and present lower yields [24]. Furthermore, plant nanovesicles present a natural diversity, varying with plant source and physio-pathological conditions. Additionally, seasonal limitations and species accessibility are important aspects that need to be considered in mass production. To overcome these limitations, plant-based biotechnology systems, such as hairy root cultures, that enable the isolation of standardized vesicles with homogenous cargo have been explored [25]. Moreover, molecular plant farming is growing in this field as plant engineering is relatively simple [6], and producing large-scale and reproducible customized EVs in vivo would completely revolutionize therapy.

Bearing all this in mind, breakthrough scientific advances in the field are expected in the next years, with innovative applications definitely being disclosed. Herein, we gather relevant information and postulate on a potential application of plant nanovesicles as the next-generation drug platform for essential oils’ delivery. The clinical use of these secondary metabolites continues to be a challenge mainly due to their high volatility and low water solubility and stability. Nanoencapsulation of essential oils in plant vesicles could represent a suitable and effective strategy to overcome these limitations and surpass deficiencies identified in other nanoencapsulation strategies such as liposomes, solid lipid nanoparticles, nano- and micro-emulsions, or polymeric nanoparticles. To avoid confusion with existing terminology, we decided to use the broader term plant nanovesicles to refer to all nanovesicles obtained from plants, disregarding their origin and isolation method.

## 2. New Applications of Essential Oils

Essential oils are a complex liquid mixture of volatile, lipophilic, and fragrant secondary metabolites obtained from aromatic plants [26]. These volatile extracts are produced and stored in secretory structures found in various plant organs, including flowers, fruits, seeds, leaves, stems, or roots. Essential oils play very relevant roles in nature, namely in plant defense and signaling processes [27], and are also explored as raw materials in the pharmaceutical, food, agronomic, cleaning, cosmetic, and perfume industries [28]. The market value of essential oils is growing and is expected to reach around USD 27 billion in the current year [29] with the essential oils from *Citrus aurantifolia*, *Citrus limon*, *Citrus sinensis*, *Cymbopogon nardus*, *Cymbopogon winterianus*, *Eucalyptus citriodora*, *Eucalyptus globulus*, *Lavandula* × *intermedia*, *Mentha canadensis*, *Mentha* × *gracilis*, *Mentha* × *piperita*, *Ocotea odorifera*, *Pogostemon cablin, Sassafras albidum,* and *Syzygium aromaticum* being the most traded, with plant production reaching over 1000 t/year [30].

To obtain these volatile extracts from plants, international guidelines recommend the use of distillation or expression in the case of *Citrus* fruits [31], thus excluding other extractive processes that resort to solvents, supercritical fluid, or are microwave-assisted. Essential oils constituents are primarily of terpenoid origin, including monoterpenes and sesquiterpenes, in the form of hydrocarbons and their oxygenated derivatives (alcohols, aldehydes, ketones, esters, ethers, peroxides, and phenols). In some cases, other compounds such as phenylpropanoids, fatty acids, and their esters, and, more rarely, nitrogen- and sulfur-containing compounds are also present [30,32].

An important aspect to consider in essential oils production is their high variability due to physiological, environmental, and genetic factors. Therefore, to ensure the high quality of commercialized oils, specific analytical guidelines published by several institutions, such as the European Pharmacopoeia, the International Standard Organization (ISO), and the World Health Organization (WHO), must be considered.

Essential oils are generally associated with antimicrobial, antiviral, and antioxidant properties [33,34,35]. Nevertheless, other bioactivities have been explored, with several studies pointing out effective anti-inflammatory, cardioprotective, neuroprotective, anti-aging, and anti-cancer effects. As several reviews are available and gather extensive and detailed information on these bioactivities [32,36,37,38,39,40,41], we only present a brief summary of the current state-of-the-art.

### 2.1. Anti-Inflammatory Potential of Essential Oils

Chronic inflammation is associated with highly prevalent diseases such as cancer, obesity, diabetes, rheumatoid arthritis, cardiovascular, respiratory, and neurodegenerative diseases, and aging. Common therapies resort to non-steroidal anti-inflammatory drugs (NSAIDs) that induce severe gastrointestinal and cardiovascular adverse effects, and although biopharmaceuticals, with safer profiles, have been used, these therapeutic agents present several limitations, including the lack of responsiveness and drug resistance, delivery problems, and production costs [42]. Therefore, plant-based therapeutic approaches have been explored, with essential oils emerging as promising, effective, and safe anti-inflammatory agents. Indeed, a search on the National Institute of Health’s online research database, Pubmed (assessed on 6 November 2022), combining ‘essential oil’ and ‘inflammation’, encountered 1779 studies. Overall, essential oils’ anti-inflammatory potential has been assessed using several in vitro and in vivo models, with the carrageenan-induced mouse paw edema model being frequently used. Several essential oils, such as clove, eucalyptus, rosemary, lavender, mint, myrrh, millefolia, and pine, have been patented and used as mixed formulations [43]. Several studies have also disclosed potential mechanisms underlying the anti-inflammatory effects of essential oils, as recently reviewed by Zhao and colleagues [41]. Overall, their anti-inflammatory mechanisms include the reduction of reactive oxygen and nitrogen species, an increase in antioxidant enzymes, and a decrease in the expression of NF-κB protein. Indeed, essential oils are able to target and inhibit multiple dysregulated signaling pathways associated with inflammation, including Toll-like receptors, nuclear transcription factor-κB, mitogen-activated protein kinases, Nod-like receptor family pyrin domain containing 3, and other auxiliary pathways (e.g., [44]).

### 2.2. Cardioprotective Potential of Essential Oils

Essential oils and their volatile compounds have shown promising effects on cardiovascular diseases and associated risk factors. A search on Pubmed (assessed on 20 September 2022), combining ‘essential oil’ and ‘cardiovascular disease’, encountered 604 studies. In general, monoterpenes such as 1,8-cineole, terpinene-4-ol, geraniol, limonene, α-pinene, thymol, and carvacrol seem to be the most studied volatile compounds showing hypotensive and anti-dyslipidemic/antidiabetic properties, whereas phenylpropanoids, such as anethol and hydroxychavicol, are very effective at avoiding platelet aggregation. Regarding essential oils, those from *Alpinia* species showed very broad effects on major risk factors associated with cardiovascular diseases and related ion channels activity; those from *Citrus* species were very effective against hypertension, and *Foeniculum vulgare* essential oil was very effective as an antidiabetic and antiplatelet agent, as already pointed out in a recent review carried out by our group [38]. Of notice, most studies do not disclose a composition–activity relation, with the activity of many compounds remaining unknown and limiting their applicability.

### 2.3. Neuroprotective and Anti-Aging Potential of Essential Oils

A relevant effect that has been associated with essential oils is their potential to exert neuroprotective and anti-aging effects, thus justifying their use in dementia, epilepsy, anxiety, and other neurological disorders such as Alzheimer’s disease (AD) and Parkinson’s disease. Indeed, currently available therapies for these disorders have limited efficacy and many times only enable symptom relief; therefore, the development of effective therapeutic strategies able to delay their progression is an urgent clinical need. Moreover, the anti-aging effects of essential oils are also relevant as they could contribute to prevent age-associated disorders. Overall, the majority of the studies carried out focus on AD as several preclinical models are available and mimic the mechanisms of action underlying this disease and behavioral features, namely memory and learning alterations [36]. Indeed, a Pubmed search on ‘essential oil’ and ‘Alzheimer’ (assessed on 20 September 2022) brings up 192 results, with the majority of the studies assessing the inhibitory potential of the extracts/compounds on acetylcholinesterase and butyrylcholinesterase, as previously stated by Rashed and colleagues [37]. Indeed, several volatile compounds such as α-pinene, δ-3-carene, 1,8-cineole, carvacrol, thymohydroquinone, α- and β-asarone, and anethole have been pointed out for their promising cholinesterase inhibitory activity [45]. In addition, many studies have shown positive effects of essential oils through anti-amyloid, antioxidant, anticholinesterase, and memory-enhancement activity, namely for *Salvia officinalis*, *Salvia lavandulifolia*, *Melissa officinalis*, *Lavandula angustifolia*, and *Rosmarinus officinalis* essential oils [46].

### 2.4. Anti-cancer Potential of Essential Oils

Regarding the anti-cancer potential of essential oils, a Pubmed search produced 1696 (assessed on 20 September 2022) results for the combined words ‘essential oil’ and ‘cancer’. The majority of the studies performed so far showed direct correlations between the eradication of tumors and cancer cell lines, with the beneficial effects of the essential oils being due to their cytotoxic, anti-proliferative, anti-tumor, or apoptotic effects as previously reviewed [40]. Indeed, several mechanisms underlying the anti-cancer effects have been reported, apoptosis being the main one [47,48]. Other mechanisms include cell cycle arrest, antimetastatic and antiangiogenic activity, increase in reactive oxygen and nitrogen species, DNA repair modulation, anti-proliferative activity, effect on tumor suppressor proteins, transcription factors, and detoxification enzymes [49].

Interestingly, analgesic, anti-inflammatory, calming, and relaxing properties are also ascribed to essential oils, rendering them adjuvant effects and alleviating cancer-related physical and emotional complaints. In addition, the combination of essential oils/volatile compounds with conventional chemotherapy has also shown promising effects and could be applied to improve cancer treatment and reduce drug toxicity, as reviewed elsewhere [40,48]. Interestingly, in this field, nanoencapsulation techniques have been largely explored, with encapsulated essential oils being able to act on and mediate target-specific drug release [40].

Despite the number of studies performed highlighting several bioactive properties, clinical trials using essential oils are sparse and additional studies are still required to assess the pharmacokinetic profile, safety, and toxicity of these compounds. Additionally, bearing in mind their high variability, standardization of the composition of essential oils and their constituents deserves special attention, and specific features such as high volatility need to be considered in order to guarantee their efficacy in a clinical setting. In this context, nanoencapsulation strategies, namely those using plant nanovesicles, emerge as promising delivery platforms, as explained in the next section.

## 3. Potential of Plant Nanovesicles for Effective Essential Oil Delivery

Essential oils have been employed in several pharmaceutical applications due to their bioactive potential. However, their conversion into dosage forms or cosmetic products poses several technological challenges, such as solubilization and storage stability issues, loss of activity due to volatile properties, and potential cytotoxicity, which may impair their performance [50]. To address these challenges, different nanoencapsulation strategies have been developed, such as emulsions, lipid nanoparticles, biopolymeric nanoparticles, clay-based nanoparticles, and inclusion complexes [51]. Although these strategies proved to prevent unwanted degradation and enhance essential oils’ efficacy with several applications already developed mainly in the food and agriculture sector [51], cytotoxicity and ecotoxicity issues still limit their clinical application. Therefore, the use of plant nanovesicles as an alternative for essential oil delivery, due to their nontoxic and nonimmunogenic character, is proposed. Among other benefits, plant nanovesicles could increase essential oils’ bioavailability and improve their chemical stability while reducing their volatility and toxicity. Additionally, by targeting these natural compounds to a specific organ, plant nanovesicles could improve their selectivity, drug delivery, efficacy, and safety, ultimately yielding a dose reduction and patient compliance and convenience improvement. Moreover, mass production could be attained with fewer costs, a major limitation when using mammalian EVs. All of this promotes the concept that plant nanovesicles are very well suited for the development of next-generation biotherapeutic and drug delivery nanoplatforms to meet the increasingly stringent requirements of current clinical challenges.

The transport and distribution of plant vesicles in vivo is the next relevant feature to take into account. Plant nanovesicles can reach distant organs and deliver their cargo by attaching to cellular membranes through receptor–ligand interactions. However, similarly to mammalian EVs, knowledge of cargo delivery is still puzzling, but it may potentially occur via the endocytic pathway, phagocytosis, micropinocytosis, or through fusion with the plasma membrane [52,53,54]. Interestingly, it seems that the cargo uptake depends on the biological context, as it can vary by plant species and recipient cell types [1]. Moreover, the type of lipids that plant vesicles contain can modify their behavior in vivo. For example, it has been shown that vesicles rich in phosphatidic acid are more retained in the intestine, while those with high amounts of phosphatidylcholine direct the migration from the intestine to the liver [55]. Moreover, it seems that vesicle biodistribution depends on their mode of administration. For example, it has been shown that intravenous injection of plant nanovesicles leads to a wide uptake by different organs, including the spleen, liver, kidney, lung, heart, and brain [56,57], while oral administration, as expected, is more directed towards the gut [55,58]. Although intravenous injection avoids first-pass hepatic metabolism, thus enabling a higher bioavailability of vesicles, PDNVs are able to resist gastric and intestinal digestion, and, therefore, oral administration seems more suitable if intestine targeting is desired. In addition, PDNVs may also penetrate the skin [59], enabling a transdermal application, and intranasal administration has also been considered, mainly when aiming for the brain [60]. This multitude of possible modes of administration anticipates the use of these vesicles in distinct pathologies. However, depending on the pathology, different administration methods would need to be optimized, as discussed in the next section.

Plant nanovesicles seem to present some targeting properties. For example, it has been demonstrated that doxorubicin can be targeted to tumors using plant nanovesicles from grapefruit [61]. Also, grapefruit-derived nanovectors loaded with methotrexate showed more specific targeting to intestinal macrophages in vivo than liposomes [15]. Aside from natural targeting specificities, surface modifications can be used to improve targeting ability or monitor vesicle delivery. For example, it was shown that functionalized mammalian EVs, with a cardiac targeting peptide, enhanced the accumulation of curcumin in the heart and increased its bioavailability [62]. Indeed, vesicle functionalization can be employed before (endogenous) or after (exogenous) vesicle isolation, as previously reviewed elsewhere [63].

Following standardized isolation, detailed characterization, and proper functionalization, plant nanovesicles could constitute a very powerful delivery platform for essential oils and their main compounds (Figure 2). Recently, Chen and colleagues provided an overview of the use of cell-derived vesicles to deliver phytochemicals [63]. Interestingly, the studies carried out so far focused on the delivery of phenolic compounds, alkaloids, and high molecular weight terpenoids such as triptolide, cucurbitacin, lycopene, and celasterol, among others. Curiously, studies on low molecular weight terpenoids such as monoterpenes and sesquiterpenes, highly prevalent in essential oils, are lacking but will certainly appear in the next years.

## 4. Current Pharmaceutical Technologies for Plant Nanovesicles Administration

Aiming to profile the requirements for the successful development of plant-derived nanocarriers, specific issues are next addressed per route of administration, followed by a compilation of the different routes of administration, advantages, and applications of essential oils encapsulation in plant nanovesicles (Table 2).

### 4.1. Oral Delivery

Essential oils have huge potential as preventive or therapeutic agents for various oral diseases, ascribed to their anti-bacterial, antifungal, and antibiofilm properties [64]. When used properly, they may prove very useful in dental therapy and may contribute to improve the quality of dental treatments. However, adverse effects, toxicity, and their interaction with other drug molecules have been evoked as reasons that prevent their wider use. Reports from the literature suggest that plant vesicles could be used in the treatment of periodontitis by inhibiting inflammation or periodontal pathogens [65]. Such effect is attributed to their involvement in the regulation of immune functions, inflammation, microbiome, and tissue regeneration, which are considered key events to be tackled in periodontitis treatment. Furthermore, plant vesicles can also be employed as drug carriers to enhance drug stability and cellular uptake in vivo [66].

Although external application in the form of gargles and mouthwashes is still the most frequent and effective way of using essential oils, they can also be taken by ingestion. In the latter case, they are generally diluted with milk, soy milk, or olive oil, as they usually have a strong aroma and flavor that can cause undesirable organoleptic properties and compromise compliance [67]. In addition, low solubility in water is one of the major limitations for essential oils’ oral administration, even if they hold a generally recognized as safe (GRAS) status. Encapsulation of essential oils in plant nanovesicles is, therefore, a potential solution to improve their bioavailability. In what concerns the stability under harsh pH and metabolic conditions of the gastrointestinal tract, there are some noticed contradictory effects. Some in vitro enzymatic digestion studies have evidenced that plant-derived vesicles are resistant to gastric and/or intestinal simulating fluids in terms of their physicochemical features (i.e., size, size distribution, surface charge) [58]. Other authors have reported an increase in size and a reversal in surface charge from positive to negative after incubation in simulated stomach solution (pH = 2) and simulated small intestine medium (pH = 6.5) of ginger-derived vesicles [68]. However, this effect could be overcome by proper enteric coating in order to retain vesicle integrity and enable the release of essential oils where they are expected to be absorbed or act locally.

Applications of plant nanovesicles as oral drug delivery systems have been directed to the treatment of colitis, bowel and liver diseases, Alzheimer’s disease, and numerous other inflammatory, neurodegenerative, and metabolic diseases. Such applications could be extended to and the therapeutic effect synergistically complemented with the essential oil administration [46,68,69]. As an example, ginger-derived nanovesicles have been developed to target intestinal epithelial cells and macrophages, preventing various inflammatory bowel diseases, such as chronic and acute colitis or colitis-related cancer, thereby promoting intestinal mucosa healing [15]. In another study, the efficacy of nine plant-derived extracellular vesicles (vegetables, fruits, spices) in inhibiting the NLRP3 inflammasome, a key regulator of the innate immune system involved in the pathogenesis of several metabolic or neurodegenerative diseases, was inspected. Ginger-derived extracellular vesicles exhibited increased targeting potential in primary macrophages that inhibited NLRP3 activation. Based on the unique features of biomolecule protection and tissue bioavailability, these vesicles were highlighted as a new class of NLRP3 inflammasome inhibitors with high potential for translational application in therapeutic modalities enrolled in the treatment of complex diseases [69].

### 4.2. Inhalation or Nasal Delivery

Inhalation provides the most rapid way to administer essential oils, prompting a fast therapeutic onset. Notwithstanding, limitations of dosages and amount of activity in inhalation administration, together with some ocular irritation as adverse effects, could hamper their continued application [70].

Encapsulation of these bioactive compounds using plant nanovesicles represents a feasible and efficient approach to modulate essential oil release throughout the respiratory tract. Increased physical stability is expected by encapsulating the essential oils, affording them protection from interactions with the environment and decreasing their volatility, therefore, sustainably enhancing their bioactivity and reducing toxicity.

Depending on the therapeutic purpose, a local or systemic effect could be retrieved. Thus, such systems could be channelled to deliver essential oils, for example, to mediate anti-inflammatory effects in allergic rhinitis patients or, alternatively, for treating mood disorders, especially depression, anxiety, and mental disorders such as sleep disorders [70,71]. Such effects may be prompted since some of them could access the brain after intranasal administration [60].

The development of these nanosystems should specifically account for the characterization of aerodynamic properties of the particulate matter and delivered dose uniformity [72].

### 4.3. Intravenous Delivery

Plant nanovesicle characteristics, especially biocompatibility, cellular uptake, and targeting capability, place these nanosystems as avant-garde nanocarrier systems for intravenous administration. Their natural and endogenic nature endow them with low potential toxicity and immunogenicity when compared with equivalent synthetic drug delivery systems, which, along with their reduced particle size and narrow distributions, enable them to remain undetected by the immune system, thereby extending the circulation period and prolonging the action time of bioactive molecules, ultimately resulting in a higher bioavailability. Additionally, their production capability in large quantities is pointed out as an advantage, being deemed critical when considering the scaling-up process [73].

As safe nanocarriers for intravenous delivery, plant nanovesicles are therefore considered promising vectors for targeted delivery in tumor diseases. For example, in a study conducted by Wang et al., grapefruit-derived nanovectors coated with membranes of activated leukocytes (IGNVs) related inflammatory receptors were found to enhance their homing ability to inflammatory tumor tissues, suggesting their potential for targeting certain cancers in the context of personalized medicine [79]. Moreover, due to their ability to cross the blood–brain barrier, they can also reach the brain, significantly expanding the delivery spectrum of essential oils [21,74,75].

### 4.4. Skin Delivery

Skin delivery stands out as the main pathway for essential oil administration. It can encompass a topical or systemic action, the latter if transdermally driven. Essential oils are often framed within the conventional profile of a typical candidate profile for dermal administration, which must have certain physicochemical characteristics to enhance permeation through the skin. These include low molecular weight (less than 500 Da), log P between 1 and 3, unionizability, adequate solubility in water (>1 mg/L), a melting point below 200 °C, and a daily dose lower than 20 mg [80]. However, given their volatility and instability against environmental factors (e.g., oxygen, light, moisture, and pH), their therapeutic effects could be compromised. Moreover, some aged essential oils, as well as oxidized terpenoids, have shown skin-sensitizing properties in addition to organoleptic and viscosity alterations, leading to a hypersensitivity reaction equivalent to allergic contact dermatitis. UV sensitivity is also an adverse effect that could cause irritation or darkening of the skin [67].

Note that the main barrier for skin permeation, the stratum corneum, consists of 10−15 nonviable layers of corneocytes composed of keratin, surrounded by a lipid matrix made of ceramides, long-chain fatty acids, triglycerides, and cholesterol [81]. Different strategies can be equated to surpass the complex structure of the stratum corneum. These may comprise passive and active approaches. The former are directed to the manipulation of the formulation, for example, by promoting an increase in the thermodynamic activity of the drug in formulations (e.g., nanocarrier systems) or by using chemical penetration enhancers able to interact with skin constituents [82,83]. The latter include the use of methodologies based on external energy (e.g., mechanical, electrical, magnetic) that act as a driving force to push the drug throughout the skin (e.g., microneedles, iontophoresis, electroporation, among others) [84,85].

As with other nanosystems, especially those structurally similar to liposomes, plant nanovesicles can be embedded in passive methods that could enhance the permeation of essential oils through the skin due to several features: (i) hydrophobic character, which confers skin resemblance; (ii) biocompatibility, which ensures a well-tolerated profile compatible with long-term use; (iii) nanometric size along with a high specific surface area that allows closer contact with the skin; (iv) high loading capacity, along with a decrease in volatile properties, which contributes to retain the essential oil and acts as a reservoir. This increases the gradient between the formulation and the stratum corneum and acts as a driving force promoting a higher absorption rate of the essential oil, resulting in greater efficacy as a delivery system. In addition, their compelling physicochemical properties underpin their modulatory role in physiological uptake processes. From a mechanism standpoint, it is hypothesized that after topical skin application, plant nanovesicles could penetrate via a trans-follicle route by a “lipid-rich channel” coating on hair follicles or intact stratum corneum, intercellular or transcellular through the lipid fusion effect, similar to what has been reported for other lipid vesicular nanosystems [76].

In addition, encapsulation of essential oils using plant nanovesicles could prevent degradation when exposed to light, oxygen, or higher temperatures (above 26 °C), improving essential oil and formulation stability [77].

Plant nanovesicles can also be combined with other inorganic nanomaterials, such as magnetic nanoparticles, to impart photothermal properties or even theranostic functions [78]. Such an approach could endow these nanosystems with enough versatility to trigger local hyperthermia, which could be useful for promoting essential oil penetration [76].

Different therapeutic effects could be envisaged when addressing topical or transdermal routes of essential oil administration, depending on the ever-stringent clinical demands, and taking advantage of their inherent anti-inflammatory, antioxidant, antimicrobial, or anti-cancer properties [67].

In addition to their therapeutic application, the use of plant nanovesicles as essential oil carriers for cosmetic purposes is also highlighted. Given the dual natural renewable source of the carrier and the essential oils, either as active ingredients or preservatives, more eco-sustainable and safer products could be produced. Indeed, several products could benefit from the advantages of encapsulating essential oils in plant nanovesicles. These range from moisturizers, lotions, and cleansers for skin care, conditioners, masks, or anti-dandruff products for hair care to lipsticks or fragrances for perfumery [86].

An enhanced cosmetic effect, given the ability to control essential oil release and counterbalance their volatility, resulting in more pleasant sensorial properties and attenuating some strong odors, sometimes unsuitable for certain cosmetic applications, a lower allergenic potential, and improved stability without the addition of chemical preservatives are pointed out as advantages that may generate increased interest in these biological nanocarriers for the cosmetics industry [86,87].

## 5. Future Prospectives and Challenges

As mentioned earlier, the use of essential oils in product formulation poses some challenges in terms of their manipulation and application, ascribed to their poor water solubility, volatility, thermal and chemical lability, inappropriate organoleptic properties, immunogenicity, and toxicological concerns. Encapsulation of essential oils in plant nanovesicles could be a very attractive option to tackle these issues. However, leveraging this environmentally sustainable strategy still requires standardized procedures to ensure the reproducibility of the process, the possibility of scale-up, in-depth physicochemical characterization to substantiate identity, and specific requirements depending on the route of administration, e.g., sterility if intravenous administration is planned, to ensure regulatory approval of these bionanosystems.

In addition, other relevant aspects need to be taken into account to enable an effective essential oil delivery, namely vesicle preparation, characterization, functionalization, stability, targeting, and delivery. Indeed, as stated above, several isolation techniques are available to obtain plant nanovesicles. Many reviews have detailed the different processes available, highlighting their pros and cons [88], and therefore, they will not be herein detailed. Overall, none of the available processes combine high yield and specificity. Moreover, as the majority of isolation techniques are adapted from protocols designed to obtain mammalian EVs, they do not account for plant specificities. For example, the presence of starch and cellulose in plant juices increases viscosity and compromises ultracentrifugation [88]. Therefore, further developments and optimizations are required, namely pretreatments, to avoid possible contamination with impurities from the isolation process. More recently, some isolation approaches have been adopted, allowing a better yield and higher purity, such as size-exclusion chromatography, field flow fractionation, and immunoaffinity enrichment [88]. Following isolation, plant nanovesicles characterization is required. The main features assessed are particle size distribution and surface charge by dynamic light scattering and nanoparticle tracking analysis, and nanovesicle morphology mainly by electron microscopy. Additionally, biochemical analysis may be performed to obtain insights into nanovesicles composition [89], and fluorescent probes can be used to track cell interaction and nanovesicles uptake by fluorescence microscopy or flow cytometry [90].

The next point to consider is loading efficiency, as the nanostructure of plant nanovesicles allows for the encapsulation of hydrophilic compounds in the core or lipophilic compounds in the surrounding lipid layer. As essential oils and their volatile compounds are lipophilic, their encapsulation would occur preferentially in the lipid layer, and both passive and active cargo loading could be used. In the first, diffusion and hydrophobic interactions between the lipid layer and the active compounds take place. This type of loading has already been used to successfully incorporate curcumin, a lipophilic compound, in mammalian EVs [91]. For active cargo loading, the lipid layer is temporarily disrupted by sonication, extrusion, or freeze–thaw cycles, thus enabling the diffusion of the active compounds before membrane integrity is restored. Although higher loading efficiencies have been reported in this case, the disruption process may modify native structures and targeting features of the vesicles. Therefore, a compromised combination of both processes, passive and active, could be a better option, as previously suggested by Akuma and colleagues [20]. Importantly, nanovesicles already contain their own cargo; therefore, content cargo offload prior to loading may be necessary to enable higher efficiency yields [20].

Stability is another key issue that determines storage time and scalability. The evaluation of the stability in biorelevant media and under storage conditions is a requirement to avoid undesirable aggregation that may negatively affect the quality and performance of the plant nanovesicles. In a study performed by Kim and colleagues, conducted to evaluate the stability of *Dendropanax morbifera* leaf-derived EVs (LEVs) alone or combined with the preservatives (3-butylene glycol or TMO), it was found that different storage temperatures and number of freeze–thaw cycles induced stability changes. Size distribution, protein content, surface charge, and cellular uptake of LEVs were altered compared to those of freshly isolated LEVs, indicating the combination LEVs-TMO as one exhibiting higher stability when stored at 4 °C [92]. However, plant nanovesicles should generally be stored under frozen conditions (−80 °C) or, alternatively, freeze-dried to preserve the integrity of the biological molecules that form their structure since the exposure to an unspecified temperature could lead to undesirable degradation due to their labile nature [18,93].

## 6. Final Remarks

The present review provides a thorough and comprehensive overview that supports the use of plant nanovesicles as innovative essential oil delivery platforms. Indeed, these volatile extracts, despite their huge preventive/therapeutic potential, continue to be disregarded in clinical practices mainly due to stability, volatility, and toxicity issues that could be surpassed by efficient delivery strategies. Plant nanovesicles have interesting carrier features and targeting properties and are known to be untaken by mammalian cells; they open very exciting avenues for future essential oil applications in the health sector. Nevertheless, standardized isolation protocols and more robust characterization procedures are needed to comply with good manufacturing practices. Moreover, despite the expected low toxicity of vesicles obtained from edible plants, further studies should be considered to better disclose plant vesicles’ pharmacokinetics, pharmacodynamics, and safety profile. Nevertheless, it seems undeniable that in the next years, the scientific knowledge in this field will continue to grow, advocating an exciting era for therapeutic plant-based nano-delivery platforms.

## Figures and Tables

**Figure 1 pharmaceutics-14-02581-f001:**
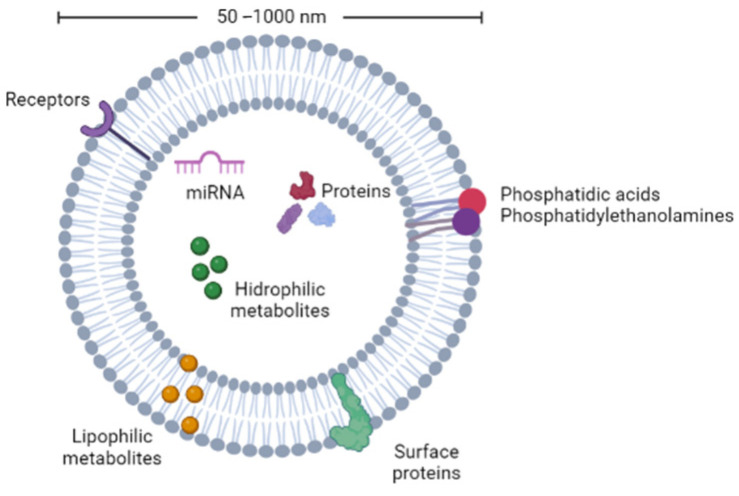
Schematic representation of biological cargo found in plant nanovesicles.

**Figure 2 pharmaceutics-14-02581-f002:**
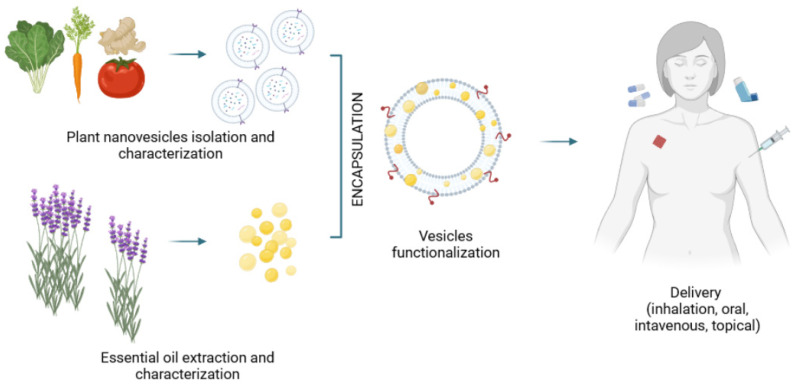
Plant-derived nanovesicles for essential oil delivery: a work-flow proposal.

**Table 1 pharmaceutics-14-02581-t001:** Patented plant nanovesicles for therapeutic application.

Invention	Plant Source	Uses	International Publication Number
A product containing plant-derived exosomes	Wheatgrass, garlic, and ginger alone or in combination	Cancer treatment and wound healing	WO/2019/027387
Plant-derived extracellular vesicle (EVs) compositions and uses thereof	Several plant families	Pro-angiogenic and anti-bacterial activity for use in the therapeutic treatment of ulcers, dermatitis, corneal damage, eye diseases, mucosal lesions, and infective lesions	WO/2020/182938
Composition for improving skin and preventing hair loss, comprising plant extract-derived extracellular vesicles	*Asparagus* plant juice	Treating hair loss or promoting hair growth	WO/2017/052267
Coated edible plant-derived microvesicle compositions and methods for using the same	Grape, grapefruit, and tomato	Inflammatory disorders, including sepsis, septic shock, colitis, colon cancer, and arthritis	WO/2015/157652

**Table 2 pharmaceutics-14-02581-t002:** Route of administration, advantages, and applications of essential oils encapsulation in plant nanovesicles.

How? Route of Administration	Why? Advantages in EO Encapsulation	Where? Applications	References
Oral delivery	-Enhanced drug stability and cellular uptake in vivo -Improved adverse effect profile -Improved organoleptic properties	-Periodontitis -Colitis, bowel, and liver diseases -Alzheimer’s disease	[15,46,64,65,66,67,68,69]
Inhalation or nasal delivery	-Essential oil release modulation throughout the respiratory tract -Decreased volatility, therefore sustainably enhancing their bioactivity and reducing toxicity	-Allergic rhinitis -Mood disorders, especially depression, anxiety, and mental disorders such as sleep disorder	[70,71,72]
Intravenous delivery	-Improved biocompatibility, cellular uptake, and targeting capability -Reduced potential toxicity and immunogenicity -Prolonged circulation period and action time of bioactives, ultimately resulting in a higher bioavailability	-Tumor diseases	[14,73,74,75]
Skin delivery	-Well-tolerated profile compatible with long-term use -High loading capacity, along with a decrease in volatile properties, contributes to retaining the essential oil and acts as a reservoir -Increased skin penetration -Improved stability	-Inflammatory conditions -Cancer -Cosmetics and toiletries	[67,76,77,78]

## Data Availability

Not applicable.

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
