# Peer review of "Plant Nanovesicles for Essential Oil Delivery"

_pharmaceutics, 2022, doi:10.3390/pharmaceutics14122581_

Round 1
Reviewer 1 Report
This is an interesting review highlighting the importance of plant nanovesicles for essential oil delivery. Authors emphasized on the systemic literature on the overview, application, targeted strategies and routes for the administration of essential oils using plant nanovesicles.
Yet, it needs minor revision in order to be accepted
Authors are suggested to recheck the MS in order to address typographical errors and grammatical syntax.
The topic is interesting and relevant at the moment, and the scientific community would value a summary of outcomes from work in the area. Therefore, graphical representations would enhance the readability of the article.
Also, authors are suggested to categorize the new applications of essential oils to enhance the seamless transition between the topics.
Author Response
Reviewer #1
Esta é uma revisão interessante destacando a importância das nanovesículas vegetais para a entrega de óleo essencial. Os autores enfatizaram na literatura sistêmica sobre o panorama, aplicação, estratégias direcionadas e rotas para a administração de óleos essenciais utilizando nanovesículas vegetais. No entanto, precisa de uma pequena revisão para ser aceito
1) Sugere-se que os autores verifiquem novamente o MS para tratar de erros tipográficos e sintaxe gramatical.
Muito obrigada pela leitura cuidada. O manuscrito foi revisto e erros tipográficos e de sintaxe gramatical corrigidos.
2) O tema é interessante e relevante no momento, e a comunidade científica valorizaria um resumo dos resultados do trabalho na área. Portanto, representações gráficas aumentariam a legibilidade do artigo.
Concordamos com a sugestão, que também foi sugerida por outro revisor, e por isso uma nova representação gráfica foi incluída.
3) Além disso, sugerem-se que os autores categorizem as novas aplicações de óleos essenciais para melhorar a transição perfeita entre os tópicos.
Mais uma vez, agradecemos a sugestão e de acordo com o sugerido foram criadas categorias para as novas bioatividades dos óleos essenciais, de forma a sistematizar melhor os dados apresentados. Foi também incluída uma nova categoria referente à atividade anti-inflamatória dos óleos essenciais, por ter sido sugerido por outro revisor.
Reviewer 2 Report
Suggested to re-write the abstract, which is not providing significant information in current form.
Several reviews are available on plant nano vesicles, however the concept of presenting concise report for essential oil delivery is an excellent idea. authors are encourage to add a graphical presentation on biological composition of plant nano-vesicles
Authors are suggested to add section on inflammations and effect of essential oil with plant derived vesicles on inflammations including mechanistic study.
Authors are suggested to a comparative characterisation of plant nano vesicles reported. More worthy if both physical, chemical, biological properties included.
Authors are also suggested to add a table indicating patented plant nano vesicles for therapeutic applications
Authors are encouraged to add extensive future prospects and challenges
Moreover suggested to add details cosmeceuticals application of essential oil fortified nano vesicles.
Please consider revision considering suggestions indicated in the attached pdf.

Author Response
Reviewer #2
1) Suggested to re-write the abstract, which is not providing significant information in current form.
We appreciate your sugestion and have modified the entire abstract to provide a more informative one.
2) Several reviews are available on plant nanovesicles, however the concept of presenting concise report for essential oil delivery is an excellent idea. Authors are encourage to add a graphical presentation on biological composition of plant nanovesicles.
Thank you for the suggestion. A graphical representation of plant’s nanovesicles cargo is provided.
3) Authors are suggested to add section on inflammations and effect of essential oil with plant derived vesicles on inflammations including mechanistic study.
We acknowledge the suggestion and have included a section reporting the anti-inflammatory potential of essential oils and their main mechanisms of action.
4) Authors are suggested to a comparative characterisation of plant nanovesicles reported. More worthy if both physical, chemical, biological properties included.
Thank you for the sugestion. Although differences between the different types of vesicles is not straightforward, we have provided information on the major differences reported.
5) Authors are also suggested to add a table indicating patented plant nanovesicles for therapeutic applications.
Thank you for the sugestion. A table with the required information was included after referring the clinical assays.
6) Authors are encouraged to add extensive future prospects and challenges.
A new section on future prospectives and challenges was added. The manuscript was reorganized as some challenges were already included in previous sections, thus avoiding unnecessary repetitions.
7) Moreover suggested to add details cosmeceuticals application of essential oil fortified nanovesicles.
Thank you for the comment. We have reorganized the modes of administration section and in the skin delivery part included information regarding cosmeceuticals application.

Reviewer 3 Report
The manuscript by Zuzarte et al. ‘Plant nanovesicles for essential oil delivery’ seems interesting and informative. The overall manuscript looks good. However, the following points should be considered before publication.
1) Here, plant nanovesicles refer to extracellular vesicles/exosomes isolated from plant sources. The term plant nanovesicles do not indicate any other nanovesicles. Thus, I suggest replacing this word with plant extracellular vesicles.
2) The authors need to discuss the isolation, loading of cargo, and characterization part for plant nanovesicles/extracellular vesicles.
3). There are several reports with plant extracellular vesicles that are used for cosmetics purposes, thus it is better to include one section for topical delivery, like cosmetics if applicable.
4) Tables, figures, or diagrams are lacking. If possible, summarize current delivery strategies in a table. Also, illustrate the schematic diagram for discussing current strategies that are employed for delivering essential oil.
5) Also, discuss the limitations/challenges of plant nanovesicles in delivering essential oil.
6) Discuss future perspective and research direction
Author Response
Reviewer #3
The manuscript by Zuzarte et al. ‘Plant nanovesicles for essential oil delivery’ seems interesting and informative. The overall manuscript looks good. However, the following points should be considered before publication.
1) Here, plant nanovesicles refer to extracellular vesicles/exosomes isolated from plant sources. The term plant nanovesicles do not indicate any other nanovesicles. Thus, I suggest replacing this word with plant extracellular vesicles.
We acknowledge your sugestion. We would like to maintain the term plant nanovesicles as we consider that it is more appropriate in the context of this manuscript. Indeed we have used the broad term plant nanovesicles on purpose to include all types of vesicles that can be obtained from plants. Many of the studies actually refer to plant-derived vesicles, obtained from blending and juicing processes, while in some cases, the term extracellular vesicles or exosomes is incorrectly applied. Herein, for essential oil delivery the origin of the vesicles is not relevant and therefore, the broad term plant nanovesicles seems, in our opinion, more appropriate. We hope you can consider this explanation and agree with our sugestion as we believe that it is quite important to draw attention to the misuse of the terminology extracellular vesicles/exosomes, frequently encountered in the literature.
2) The authors need to discuss the isolation, loading of cargo, and characterization part for plant nanovesicles/extracellular vesicles.
A better characterization of the referred points is presented.
3). There are several reports with plant extracellular vesicles that are used for cosmetics purposes, thus it is better to include one section for topical delivery, like cosmetics if applicable.
Thank you for the comment. We have reorganized the modes of administration section and in the skin delivery part included information regarding cosmeceuticals application.
4) Tables, figures, or diagrams are lacking. If possible, summarize current delivery strategies in a table. Also, illustrate the schematic diagram for discussing current strategies that are employed for delivering essential oil.
We appreciate the suggestions and have compiled the current delivery strategies in a table. The schematic representation is also shown in figure 2.
5) Also, discuss the limitations/challenges of plant nanovesicles in delivering essential oil.
This information was included in the new section on future prospectives and challenges.
6) Discuss future perspective and research direction
A new section on future prospectives and challenges was added.

Round 2
Reviewer 2 Report
The manuscript can be accepted in current formate, authors have reflected required correction suggested.
Reviewer 3 Report
Authors have edited the manuscript and added as per my comments and suggestions. Now, the manuscript looks far better. Thus, I recommend for publication.